# A Gang of Bandits

**Nicolò Cesa-Bianchi**
Università degli Studi di Milano, Italy
`nicolo.cesa-bianchi@unimi.it`

**Claudio Gentile**
University of Insubria, Italy
`claudio.gentile@uninsubria.it`

**Giovanni Zappella**
Università degli Studi di Milano, Italy
`giovanni.zappella@unimi.it`

## Abstract

Multi-armed bandit problems formalize the exploration-exploitation trade-offs arising in several industrially relevant applications, such as online advertisement and, more generally, recommendation systems. In many cases, however, these applications have a strong social component, whose integration in the bandit algorithm could lead to a dramatic performance increase. For instance, content may be served to a group of users by taking advantage of an underlying network of social relationships among them. In this paper, we introduce novel algorithmic approaches to the solution of such networked bandit problems. More specifically, we design and analyze a global recommendation strategy which allocates a bandit algorithm to each network node (user) and allows it to "share" signals (contexts and payoffs) with the neghboring nodes. We then derive two more scalable variants of this strategy based on different ways of clustering the graph nodes. We experimentally compare the algorithm and its variants to state-of-the-art methods for contextual bandits that do not use the relational information. Our experiments, carried out on synthetic and real-world datasets, show a consistent increase in prediction performance obtained by exploiting the network structure.

## 1  Introduction

The ability of a website to present personalized content recommendations is playing an increasingly crucial role in achieving user satisfaction. Because of the appearance of new content, and due to the ever-changing nature of content popularity, modern approaches to content recommendation are strongly adaptive, and attempt to match as closely as possible users' interests by learning good mappings between available content and users. These mappings are based on "contexts", that is sets of features that, typically, are extracted from both contents and users. The need to focus on content that raises the user interest and, simultaneously, the need of exploring new content in order to globally improve the user experience creates an exploration-exploitation dilemma, which is commonly formalized as a multi-armed bandit problem. Indeed, contextual bandits have become a reference model for the study of adaptive techniques in recommender systems (e.g, [5, 7, 15] ). In many cases, however, the users targeted by a recommender system form a social network. The network structure provides an important additional source of information, revealing potential affinities between pairs of users. The exploitation of such affinities could lead to a dramatic increase in the quality of the recommendations. This is because the knowledge gathered about the interests of a given user may be exploited to improve the recommendation to the user's friends. In this work, an algorithmic approach to networked contextual bandits is proposed which is provably able to leverage user similarities represented as a graph. Our approach consists in running an instance of a contextual bandit algorithm at each network node. These instances are allowed to interact during the learning process,

sharing contexts and user feedbacks. Under the modeling assumption that user similarities are properly reflected by the network structure, interactions allow to effectively speed up the learning process that takes place at each node. This mechanism is implemented by running instances of a linear contextual bandit algorithm in a specific reproducing kernel Hilbert space (RKHS). The underlying kernel, previously used for solving online multitask classification problems (e.g., [8]), is defined in terms of the Laplacian matrix of the graph. The Laplacian matrix provides the information we rely upon to share user feedbacks from one node to the others, according to the network structure. Since the Laplacian kernel is linear, the implementation in kernel space is straightforward. Moreover, the existing performance guarantees for the specific bandit algorithm we use can be directly lifted to the RKHS, and expressed in terms of spectral properties of the user network. Despite its crispness, the principled approach described above has two drawbacks hindering its practical usage. First, running a network of linear contextual bandit algorithms with a Laplacian-based feedback sharing mechanism may cause significant scaling problems, even on small to medium sized social networks. Second, the social information provided by the network structure at hand need not be fully reliable in accounting for user behavior similarities. Clearly enough, the more such algorithms hinge on the network to improve learning rates, the more they are penalized if the network information is noisy and/or misleading. After collecting empirical evidence on the sensitivity of networked bandit methods to graph noise, we propose two simple modifications to our basic strategy, both aimed at circumventing the above issues by clustering the graph nodes. The first approach reduces graph noise simply by deleting edges between pairs of clusters. By doing that, we end up running a scaled down independent instance of our original strategy on each cluster. The second approach treats each cluster as a single node of a much smaller cluster network. In both cases, we are able to empirically improve prediction performance, and simultaneously achieve dramatic savings in running times. We run experiments on two real-world datasets: one is extracted from the social bookmarking web service Delicious, and the other one from the music streaming platform Last.fm.

## 2 Related work

The benefit of using social relationships in order to improve the quality of recommendations is a recognized fact in the literature of content recommender systems —see e.g., [5, 13, 18] and the survey [3]. Linear models for contextual bandits were introduced in [4]. Their application to personalized content recommendation was pioneered in [15], where the LinUCB algorithm was introduced. An analysis of LinUCB was provided in the subsequent work [9]. To the best of our knowledge, this is the first work that combines contextual bandits with the social graph information. However, non-contextual stochastic bandits in social networks were studied in a recent independent work [20]. Other works, such as [2, 19], consider contextual bandits assuming metric or probabilistic dependencies on the product space of contexts and actions. A different viewpoint, where each action reveals information about other actions' payoffs, is the one studied in [7, 16], though without the context provided by feature vectors. A non-contextual model of bandit algorithms running on the nodes of a graph was studied in [14]. In that work, only one node reveals its payoffs, and the statistical information acquired by this node over time is spread across the entire network following the graphical structure. The main result shows that the information flow rate is sufficient to control regret at each node of the network. More recently, a new model of distributed non-contextual bandit algorithms has been presented in [21], where the number of communications among the nodes is limited, and all the nodes in the network have the same best action.

## 3 Learning model

We assume the social relationships over users are encoded as a *known* undirected and connected graph $G = (V, E)$, where $V = \{1, \ldots, n\}$ represents a set of $n$ users, and the edges in $E$ represent the social links over pairs of users. Recall that a graph $G$ can be equivalently defined in terms of its Laplacian matrix $L = \left[L_{i,j}\right]_{i,j=1}^{n}$, where $L_{i,i}$ is the degree of node $i$ (i.e., the number of incoming/outgoing edges) and, for $i \neq j$, $L_{i,j}$ equals $-1$ if $(i, j) \in E$, and 0 otherwise. Learning proceeds in a sequential fashion: At each time step $t = 1, 2, \ldots$, the learner receives a user index $i_t \in V$ together with a set of context vectors $C_{i_t} = \{\boldsymbol{x}_{t,1}, \boldsymbol{x}_{t,2}, \ldots, \boldsymbol{x}_{t,c_t}\} \subseteq \mathbb{R}^d$. The learner then selects some $k_t \in C_{i_t}$ to recommend to user $i_t$ and observes some payoff $a_t \in [-1, 1]$, a function

of $i_t$ and $\bar{\boldsymbol{x}}_t = \boldsymbol{x}_{t,k_t}$. No assumptions whatsoever are made on the way index $i_t$ and set $C_{i_t}$ are generated, in that they can arbitrarily depend on past choices made by the algorithm.[1]

A standard modeling assumption for bandit problems with contextual information (one that is also adopted here) is to assume that rewards are generated by noisy versions of unknown linear functions of the context vectors. That is, we assume each node $i \in V$ hosts an unknown parameter vector $\boldsymbol{u}_i \in \mathbb{R}^d$, and that the reward value $a_i(\boldsymbol{x})$ associated with node $i$ and context vector $\boldsymbol{x} \in \mathbb{R}^d$ is given by the random variable $a_i(\boldsymbol{x}) = \boldsymbol{u}_i^\top \boldsymbol{x} + \epsilon_i(\boldsymbol{x})$, where $\epsilon_i(\boldsymbol{x})$ is a conditionally zero-mean and bounded variance noise term. Specifically, denoting by $\mathbb{E}_t[\,\cdot\,]$ the conditional expectation $\mathbb{E}\big[\,\cdot\,\big|\,(i_1, C_{i_1}, a_1), \ldots, (i_{t-1}, C_{i_{t-1}}, a_{t-1})\,\big]$, we take the general approach of [1], and assume that for any fixed $i \in V$ and $\boldsymbol{x} \in \mathbb{R}^d$, the variable $\epsilon_i(\boldsymbol{x})$ is conditionally sub-Gaussian with variance parameter $\sigma^2 > 0$, namely, $\mathbb{E}_t\big[\exp(\gamma\,\epsilon_i(\boldsymbol{x}))\big] \leq \exp\big(\sigma^2\,\gamma^2/2\big)$ for all $\gamma \in \mathbb{R}$ and all $\boldsymbol{x}, i$. This implies $\mathbb{E}_t[\epsilon_i(\boldsymbol{x})] = 0$ and $\mathbb{V}_t\big[\epsilon_i(\boldsymbol{x})\big] \leq \sigma^2$, where $\mathbb{V}_t[\cdot]$ is a shorthand for the conditional variance $\mathbb{V}\big[\,\cdot\,\big|\,(i_1, C_{i_1}, a_1), \ldots, (i_{t-1}, C_{i_{t-1}}, a_{t-1})\,\big]$. So we clearly have $\mathbb{E}_t[a_i(\boldsymbol{x})] = \boldsymbol{u}_i^\top \boldsymbol{x}$ and $\mathbb{V}_t\big[a_i(\boldsymbol{x})\big] \leq \sigma^2$. Therefore, $\boldsymbol{u}_i^\top \boldsymbol{x}$ is the expected reward observed at node $i$ for context vector $\boldsymbol{x}$. In the special case when the noise $\epsilon_i(\boldsymbol{x})$ is a bounded random variable taking values in the range $[-1, 1]$, this implies $\mathbb{V}_t[a_i(\boldsymbol{x})] \leq 4$.

The regret $r_t$ of the learner at time $t$ is the amount by which the average reward of the best choice in hindsight at node $i_t$ exceeds the average reward of the algorithm's choice, i.e.,

$$r_t = \left( \max_{\boldsymbol{x} \in C_{i_t}} \boldsymbol{u}_{i_t}^\top \boldsymbol{x} \right) - \boldsymbol{u}_{i_t}^\top \bar{\boldsymbol{x}}_t \;.$$

The goal of the algorithm is to bound with high probability (over the noise variables $\epsilon_{i_t}$) the cumulative regret $\sum_{t=1}^T r_t$ for the given sequence of nodes $i_1, \ldots, i_T$ and observed context vector sets $C_{i_1}, \ldots, C_{i_T}$. We model the similarity among users in $V$ by making the assumption that nearby users hold similar underlying vectors $\boldsymbol{u}_i$, so that reward signals received at a given node $i_t$ at time $t$ are also, to some extent, informative to learn the behavior of other users $j$ connected to $i_t$ within $G$. We make this more precise by taking the perspective of known multitask learning settings (e.g., [8]), and assume that

$$\sum_{(i,j) \in E} \|\boldsymbol{u}_i - \boldsymbol{u}_j\|^2 \tag{1}$$

is small compared to $\sum_{i \in V} \|\boldsymbol{u}_i\|^2$, where $\|\cdot\|$ denotes the standard Euclidean norm of vectors. That is, although (1) may possibly contain a quadratic number of terms, the closeness of vectors lying on adjacent nodes in $G$ makes this sum comparatively smaller than the actual length of such vectors. This will be our working assumption throughout, one that motivates the Laplacian-regularized algorithm presented in Section 4, and empirically tested in Section 5.

## 4 Algorithm and regret analysis

Our bandit algorithm maintains at time $t$ an estimate $\boldsymbol{w}_{i,t}$ for vector $\boldsymbol{u}_i$. Vectors $\boldsymbol{w}_{i,t}$ are updated based on the reward signals as in a standard linear bandit algorithm (e.g., [9]) operating on the context vectors contained in $C_{i_t}$. Every node $i$ of $G$ hosts a linear bandit algorithm like the one described in Figure 1. The algorithm in Figure 1 maintains at time $t$ a prototype vector $\boldsymbol{w}_t$ which is the result of a standard linear least-squares approximation to the unknown parameter vector $\boldsymbol{u}$ associated with the node under consideration. In particular, $\boldsymbol{w}_{t-1}$ is obtained by multiplying the inverse correlation matrix $M_{t-1}$ and the bias vector $\boldsymbol{b}_{t-1}$. At each time $t = 1, 2, \ldots$, the algorithm receives context vectors $\boldsymbol{x}_{t,1}, \ldots, \boldsymbol{x}_{t,c_t}$ contained in $C_t$, and must select one among them. The linear bandit algorithm selects $\bar{\boldsymbol{x}}_t = \boldsymbol{x}_{t,k_t}$ as the vector in $C_t$ that maximizes an upper-confidence-corrected estimation of the expected reward achieved over context vectors $\boldsymbol{x}_{t,k}$. The estimation is based on the current $\boldsymbol{w}_{t-1}$, while the upper confidence level $\text{CB}_t$ is suggested by the standard analysis of linear bandit algorithms —see, e.g., [1, 9, 10]. Once the actual reward $a_t$ associated with $\bar{\boldsymbol{x}}_t$ is observed, the algorithm uses $\bar{\boldsymbol{x}}_t$ for updating $M_{t-1}$ to $M_t$ via a rank-one adjustment, and $\boldsymbol{b}_{t-1}$ to $\boldsymbol{b}_t$ via an additive update whose learning rate is precisely $a_t$. This algorithm can be seen as a version of LinUCB [9], a linear bandit algorithm derived from LinRel [4].

**Init**: $\boldsymbol{b}_0 = \mathbf{0} \in \mathbb{R}^d$ and $M_0 = I \in \mathbb{R}^{d \times d}$;
**for** $t = 1, 2, \ldots, T$ **do**
    Set $\boldsymbol{w}_{t-1} = M_{t-1}^{-1} \boldsymbol{b}_{t-1}$;
    Get context $C_t = \{\boldsymbol{x}_{t,1}, \ldots, \boldsymbol{x}_{t,c_t}\}$;
    Set
$$k_t = \operatorname*{argmax}_{k=1,\ldots,c_t} \left( \boldsymbol{w}_{t-1}^\top \boldsymbol{x}_{t,k} + \mathrm{CB}_t(\boldsymbol{x}_{t,k}) \right)$$
    where
$$\mathrm{CB}_t(\boldsymbol{x}_{t,k}) = \sqrt{\boldsymbol{x}_{t,k}^\top M_{t-1}^{-1} \boldsymbol{x}_{t,k}} \left( \sigma \sqrt{\ln \frac{|M_t|}{\delta}} + \|\boldsymbol{u}\| \right)$$
    Set $\bar{\boldsymbol{x}}_t = \boldsymbol{x}_{t,k_t}$;
    Observe reward $a_t \in [-1, 1]$;
    Update
        • $M_t = M_{t-1} + \bar{\boldsymbol{x}}_t \bar{\boldsymbol{x}}_t^\top$,
        • $\boldsymbol{b}_t = \boldsymbol{b}_{t-1} + a_t \bar{\boldsymbol{x}}_t$.
**end for**

Figure 1: Pseudocode of the linear bandit algorithm sitting at each node $i$ of the given graph.

**Init**: $\boldsymbol{b}_0 = \mathbf{0} \in \mathbb{R}^{dn}$ and $M_0 = I \in \mathbb{R}^{dn \times dn}$;
**for** $t = 1, 2, \ldots, T$ **do**
    Set $\boldsymbol{w}_{t-1} = M_{t-1}^{-1} \boldsymbol{b}_{t-1}$;
    Get $i_t \in V$, context $C_{i_t} = \{\boldsymbol{x}_{t,1}, \ldots, \boldsymbol{x}_{t,c_t}\}$;
    Construct vectors $\boldsymbol{\phi}_{i_t}(\boldsymbol{x}_{t,1}), \ldots, \boldsymbol{\phi}_{i_t}(\boldsymbol{x}_{t,c_t})$, and modified vectors $\widetilde{\boldsymbol{\phi}}_{t,1}, \ldots, \widetilde{\boldsymbol{\phi}}_{t,c_t}$, where
$$\widetilde{\boldsymbol{\phi}}_{t,k} = A_\otimes^{-1/2} \boldsymbol{\phi}_{i_t}(\boldsymbol{x}_{t,k}), \quad k = 1, \ldots, c_t;$$
    Set $k_t = \operatorname*{argmax}\limits_{k=1,\ldots,c_t} \left( \boldsymbol{w}_{t-1}^\top \widetilde{\boldsymbol{\phi}}_{t,k} + \mathrm{CB}_t(\widetilde{\boldsymbol{\phi}}_{t,k}) \right)$ where
$$\mathrm{CB}_t(\widetilde{\boldsymbol{\phi}}_{t,k}) = \sqrt{\widetilde{\boldsymbol{\phi}}_{t,k}^\top M_{t-1}^{-1} \widetilde{\boldsymbol{\phi}}_{t,k}} \left( \sigma \sqrt{\ln \frac{|M_t|}{\delta}} + \|\widetilde{U}\| \right)$$
    Observe reward $a_t \in [-1, 1]$ at node $i_t$;
    Update
        • $M_t = M_{t-1} + \widetilde{\boldsymbol{\phi}}_{t,k_t} \widetilde{\boldsymbol{\phi}}_{t,k_t}^\top$,
        • $\boldsymbol{b}_t = \boldsymbol{b}_{t-1} + a_t \widetilde{\boldsymbol{\phi}}_{t,k}$.
**end for**

Figure 2: Pseudocode of the GOB.Lin algorithm.

We now turn to describing our **GOB.Lin** (Gang Of Bandits, Linear version) algorithm. GOB.Lin lets the algorithm in Figure 1 operate on each node $i$ of $G$ (we should then add subscript $i$ throughout, replacing $\boldsymbol{w}_t$ by $\boldsymbol{w}_{i,t}$, $M_t$ by $M_{i,t}$, and so forth). The updates $M_{i,t-1} \to M_{i,t}$ and $\boldsymbol{b}_{i,t-1} \to \boldsymbol{b}_{i,t}$ are performed at node $i$ through vector $\bar{\boldsymbol{x}}_t$ both when $i = i_t$ (i.e., when node $i$ is the one which the context vectors in $C_{i_t}$ refer to) and to a lesser extent when $i \neq i_t$ (i.e., when node $i$ is not the one which the vectors in $C_{i_t}$ refer to). This is because, as we said, the payoff $a_t$ received for node $i_t$ is somehow informative also for all other nodes $i \neq i_t$. In other words, because we are assuming the underlying parameter vectors $\boldsymbol{u}_i$ are close to each other, we should let the corresponding prototype vectors $\boldsymbol{w}_{i,t}$ undergo similar updates, so as to also keep the $\boldsymbol{w}_{i,t}$ close to each other over time.

With this in mind, we now describe GOB.Lin in more detail. It is convenient to introduce first some extra matrix notation. Let $A = I_n + L$, where $L$ is the Laplacian matrix associated with $G$, and $I_n$ is the $n \times n$ identity matrix. Set $A_\otimes = A \otimes I_d$, the Kronecker product[2] of matrices $A$ and $I_d$. Moreover, the "compound" descriptor for the pairing $(i, \boldsymbol{x})$ is given by the long (and sparse) vector $\boldsymbol{\phi}_i(\boldsymbol{x}) \in \mathbb{R}^{dn}$ defined as

$$\boldsymbol{\phi}_i(\boldsymbol{x})^\top = ( \underbrace{0, \ldots, 0}_{(i-1)d \text{ times}}, \boldsymbol{x}^\top, \underbrace{0, \ldots, 0}_{(n-i)d \text{ times}} ) .$$

With the above notation handy, a compact description of GOB.Lin is presented in Figure 2, where we deliberately tried to mimic the pseudocode of Figure 1. Notice that in Figure 2 we overloaded the notation for the confidence bound $\mathrm{CB}_t$, which is now defined in terms of the Laplacian $L$ of $G$. In particular, $\|\boldsymbol{u}\|$ in Figure 1 is replaced in Figure 2 by $\|\widetilde{U}\|$, where $\widetilde{U} = A_\otimes^{1/2} U$ and we define $U = (\boldsymbol{u}_1^\top, \boldsymbol{u}_2^\top, \ldots, \boldsymbol{u}_n^\top)^\top \in \mathbb{R}^{dn}$. Clearly enough, the potentially unknown quantities $\|\boldsymbol{u}\|$ and $\|\widetilde{U}\|$ in the two expressions for $\mathrm{CB}_t$ can be replaced by suitable upper bounds.

We now explain how the modified long vectors $\widetilde{\boldsymbol{\phi}}_{t,k} = A_\otimes^{-1/2} \boldsymbol{\phi}_{i_t}(\boldsymbol{x}_{t,k})$ act in the update of matrix $M_t$ and vector $\boldsymbol{b}_t$. First, observe that if $A_\otimes$ were the identity matrix then, according to how the long vectors $\boldsymbol{\phi}_{i_t}(\boldsymbol{x}_{t,k})$ are defined, $M_t$ would be a block-diagonal matrix $M_t = \mathrm{diag}(D_1, \ldots, D_n)$, whose $i$-th block $D_i$ is the $d \times d$ matrix $D_i = I_d + \sum_{t : k_t = i} \boldsymbol{x}_t \boldsymbol{x}_t^\top$. Similarly, $\boldsymbol{b}_t$ would be the $dn$-long vector whose $i$-th $d$-dimensional block contains $\sum_{t : k_t = i} a_t \boldsymbol{x}_t$. This would be equivalent to running $n$ independent linear bandit algorithms (Figure 1), one per node of $G$. Now, because $A_\otimes$ is not the identity, but contains graph $G$ represented through its Laplacian matrix, the selected vector $\boldsymbol{x}_{t,k_t} \in C_{i_t}$ for node $i_t$ gets spread via $A_\otimes^{-1/2}$ from the $i_t$-th block over all other blocks, thereby making the contextual information contained in $\boldsymbol{x}_{t,k_t}$ available to update the internal status

of all other nodes. Yet, the only reward signal observed at time $t$ is the one available at node $i_t$. A theoretical analysis of GOB.Lin relying on the learning model of Section 3 is sketched in Section 4.1.

GOB.Lin's running time is mainly affected by the inversion of the $dn \times dn$ matrix $M_t$, which can be performed in time of order $(dn)^2$ per round by using well-known formulas for incremental matrix inversions. The same quadratic dependence holds for memory requirements. In our experiments, we observed that projecting the contexts on the principal components improved performance. Hence, the quadratic dependence on the context vector dimension $d$ is not really hurting us in practice. On the other hand, the quadratic dependence on the number of nodes $n$ may be a significant limitation to GOB.Lin's practical deployment. In the next section, we show that simple graph compression schemes (like node clustering) can conveniently be applied to both reduce edge noise and bring the algorithm to reasonable scaling behaviors.

## 4.1 Regret Analysis

We now provide a regret analysis for GOB.Lin that relies on the high probability analysis contained in [1] (Theorem 2 therein). The analysis can be seen as a combination of the multitask kernel contained in, e.g., [8, 17, 12] and a version of the linear bandit algorithm described and analyzed in [1].

**Theorem 1.** *Let the GOB.Lin algorithm of Figure 2 be run on graph* $G = (V, E)$, $V = \{1, \ldots, n\}$, *hosting at each node* $i \in V$ *vector* $\boldsymbol{u}_i \in \mathbb{R}^d$. *Define*

$$L(\boldsymbol{u}_1, \ldots, \boldsymbol{u}_n) = \sum_{i \in V} \|\boldsymbol{u}_i\|^2 + \sum_{(i,j) \in E} \|\boldsymbol{u}_i - \boldsymbol{u}_j\|^2 .$$

*Let also the sequence of context vectors* $\boldsymbol{x}_{t,k}$ *be such that* $\|\boldsymbol{x}_{t,k}\| \leq B$, *for all* $k = 1, \ldots, c_t$, *and* $t = 1, \ldots, T$. *Then the cumulative regret satisfies*

$$\sum_{t=1}^T r_t \leq 2 \sqrt{T \left( 2\sigma^2 \ln \frac{|M_T|}{\delta} + 2L(\boldsymbol{u}_1, \ldots, \boldsymbol{u}_n) \right) (1 + B^2) \ln |M_T|}$$

*with probability at least* $1 - \delta$.

Compared to running $n$ independent bandit algorithms (which corresponds to $A_\otimes$ being the identity matrix), the bound in the above theorem has an extra term $\sum_{(i,j) \in E} \|\boldsymbol{u}_i - \boldsymbol{u}_j\|^2$, which we assume small according to our working assumption. However, the bound has also a significantly smaller log determinant $\ln |M_T|$ on the resulting matrix $M_T$, due to the construction of $\widetilde{\phi}_{t,k}$ via $A_\otimes^{-1/2}$. In particular, when the graph is very dense, the log determinant in GOB.Lin is a factor $n$ smaller than the corresponding term for the $n$ independent bandit case (see, e.g.,[8], Section 4.2 therein). To make things clear, consider two extreme situations. When $G$ has no edges then $\mathrm{TR}(M_T) = \mathrm{TR}(I) + T = nd + T$, hence $\ln |M_T| \leq dn \ln(1 + T/(dn))$. On the other hand, When $G$ is the complete graph then $\mathrm{TR}(M_T) = \mathrm{TR}(I) + 2t/(n+1) = nd + 2T/(n+1)$, hence $\ln |M_T| \leq dn \ln(1 + 2T/(dn(n+1)))$. The exact behavior of $\ln |M_t|$ (one that would ensure a significant advantage in practice) depends on the actual interplay between the data and the graph, so that the above linear dependence on $dn$ is really a coarse upper bound.

## 5 Experiments

In this section, we present an empirical comparison of GOB.Lin (and its variants) to linear bandit algorithms which do not exploit the relational information provided by the graph. We run our experiments by approximating the $\mathrm{CB}_t$ function in Figure 1 with the simplified expression $\alpha \sqrt{\boldsymbol{x}_{t,k}^\top M_{t-1}^{-1} \boldsymbol{x}_{t,k} \log(t+1)}$, and the $\mathrm{CB}_t$ function in Figure 2 with the corresponding expression in which $\boldsymbol{x}_{t,k}$ is replaced by $\widetilde{\phi}_{t,k}$. In both cases, the factor $\alpha$ is used as tunable parameter. Our preliminary experiments show that this approximation does not affect the predictive performances of the algorithms, while it speeds up computation significantly. We tested our algorithm and its competitors on a synthetic dataset and two freely available real-world datasets extracted from the social bookmarking web service Delicious and from the music streaming service Last.fm. These datasets are structured as follows.

**4Cliques.** This is an artificial dataset whose graph contains four cliques of 25 nodes each to which we added graph noise. This noise consists in picking a random pair of nodes and deleting or creating an edge between them. More precisely, we created a $n \times n$ symmetric noise matrix of random numbers in $[0, 1]$, and we selected a threshold value such that the expected number of matrix elements above this value is exactly some chosen noise rate parameter. Then we set to $1$ all the entries whose content is above the threshold, and to zero the remaining ones. Finally, we XORed the noise matrix with the graph adjacency matrix, thus obtaining a noisy version of the original graph.

**Last.fm.** This is a social network containing 1,892 nodes and 12,717 edges. There are 17,632 items (artists), described by 11,946 tags. The dataset contains information about the listened artists, and we used this information in order to create the payoffs: if a user listened to an artist at least once the payoff is $1$, otherwise the payoff is $0$.

**Delicious.** This is a network with 1,861 nodes and 7,668 edges. There are 69,226 items (URLs) described by 53,388 tags. The payoffs were created using the information about the bookmarked URLs for each user: the payoff is $1$ if the user bookmarked the URL, otherwise the payoff is $0$.

Last.fm and Delicious were created by the Information Retrieval group at Universidad Autonoma de Madrid for the HetRec 2011 Workshop [6] with the goal of investigating the usage of heterogeneous information in recommendation systems.[3] These two networks are structurally different: on Delicious, payoffs depend on users more strongly than on Last.fm. In other words, there are more popular artists, whom everybody listens to, than popular websites, which everybody bookmarks — see Figure 3. This makes a huge difference in practice, and the choice of these two datasets allow us to make a more realistic comparison of recommendation techniques. Since we did not remove any items from these datasets (neither the most frequent nor the least frequent), these differences do influence the behavior of all algorithms —see below.

Some statistics about Last.fm and Delicious are reported in Table 1. In Figure 3 we plotted the distribution of the number of preferences per item in order to make clear and visible the differences explained in the previous paragraphs.[4]

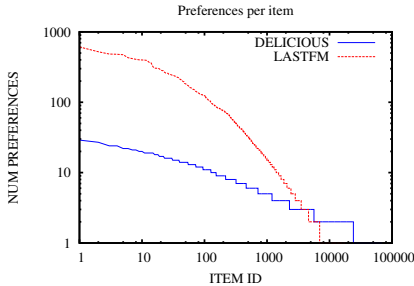

Figure 3: Plot of the number of preferences per item (users who bookmarked the URL or listened to an artist). Both axes have logarithmic scale.

|  | LAST.FM | DELICIOUS |
|---|---|---|
| NODES | 1892 | 1867 |
| EDGES | 12717 | 7668 |
| AVG. DEGREE | 13.443 | 8.21 |
| ITEMS | 17632 | 69226 |
| NONZERO PAYOFFS | 92834 | 104799 |
| TAGS | 11946 | 53388 |

Table 1: Main statistics for Last.fm and Delicious. ITEMS counts the overall number of items, across all users, from which $C_t$ is selected. NONZERO PAYOFFS is the number of pairs (user, item) for which we have a nonzero payoff. TAGS is the number of distinct tags that were used to describe the items.

We preprocessed datasets by breaking down the tags into smaller tags made up of single words. In fact, many users tend to create tags like "webdesign_tutorial_css". This tag has been splitted into three smaller tags corresponding to the three words therein. More generally, we splitted all compound tags containing underscores, hyphens and apexes. This makes sense because users create tags independently, and we may have both "rock_and_roll" and "rock_n_roll". Because of this splitting operation, the number of unique tags decreased from 11,946 to 6,036 on Last.fm and from 53,388 to 9,949 on Delicious. On Delicious, we also removed all tags occurring less than ten times.[5] The

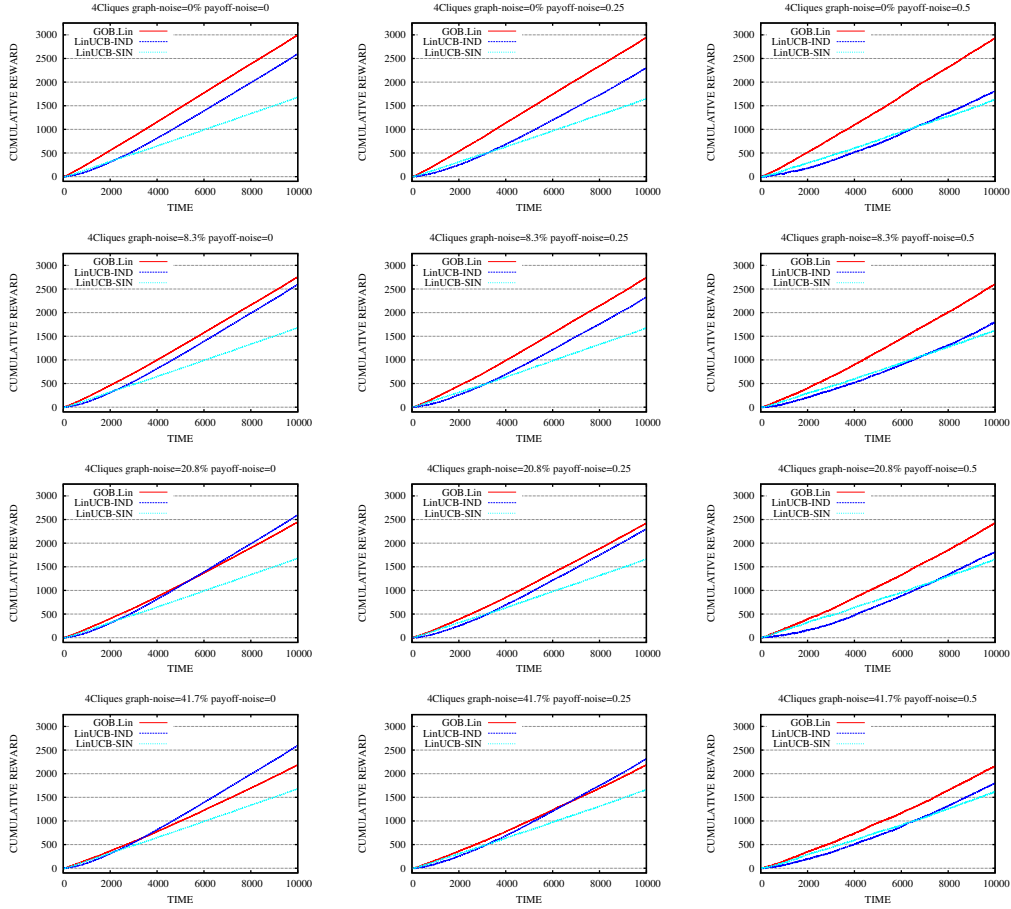

Table 2: Normalized cumulated reward for different levels of graph noise (expected fraction of perturbed edges) and payoff noise (largest absolute value of noise term $\epsilon$) on the 4Cliques dataset. Graph noise increases from top to bottom, payoff noise increases from left to right. GOB.Lin is clearly more robust to payoff noise than its competitors. On the other hand, GOB.Lin is sensitive to high levels of graph noise. In the last row, graph noise is $41.7\%$, i.e., the number of perturbed edges is 500 out of 1200 edges of the original graph.

algorithms we tested do not use any prior information about which user provided a specific tag. We used all tags associated with a single item to create a TF-IDF context vector that uniquely represents that item, independent of which user the item is proposed to. In both datasets, we only retained the first 25 principal components of context vectors, so that $\boldsymbol{x}_{t,k} \in \mathbb{R}^{25}$ for all $t$ and $k$. We generated random context sets $C_{i_t}$ of size 25 for Last.fm and Delicious, and of size 10 for 4Cliques. In practical scenarios, these numbers would be varying over time, but we kept them fixed so as to simplify the experimental setting. In 4Cliques we assigned the same unit norm random vector $\boldsymbol{u}_i$ to every node in the same clique $i$ of the original graph (before adding graph noise). Payoffs were then generated according to the following stochastic model: $a_i(\boldsymbol{x}) = \boldsymbol{u}_i^\top \boldsymbol{x} + \epsilon$, where $\epsilon$ (the payoff noise) is uniformly distributed in a bounded interval centered around zero. For Delicious and Last.fm, we created a set of context vectors for every round $t$ as follows: we first picked $i_t$ uniformly at random in $\{1, \ldots, n\}$. Then, we generated context vectors $\boldsymbol{x}_{t,1}, \ldots, \boldsymbol{x}_{t,25}$ in $C_{i_t}$ by picking 24 vectors at random from the dataset and one among those vectors with nonzero payoff for user $i_t$. This is necessary in order to avoid a meaningless comparison: with high probability, a purely random selection would result in payoffs equal to zero for all the context vectors in $C_{i_t}$. In our experimental comparison, we tested GOB.Lin and its variants against two baselines: a baseline LinUCB-IND that runs an independent instance of the algorithm in Figure 1 at each node (this is equivalent to running GOB.Lin in Figure 2 with $A_\otimes = I_{dn}$) and a baseline LinUCB-SIN, which runs a single instance of the algorithm in Figure 1 shared by all the nodes. LinUCB-IND turns to be

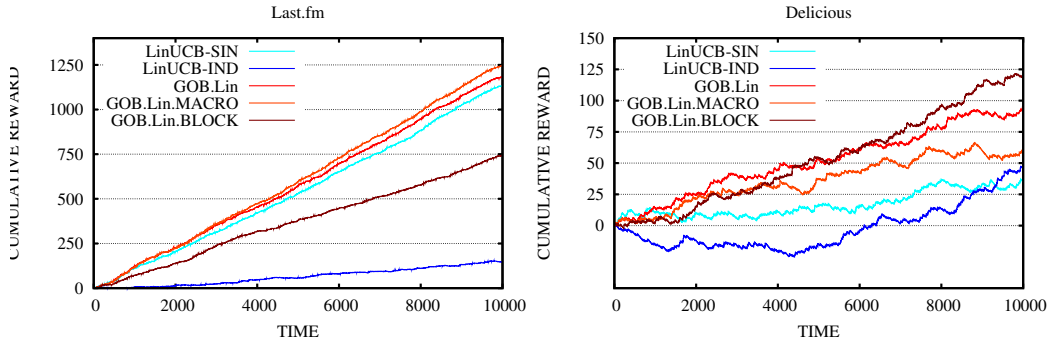

Figure 4: Cumulative reward for all the bandit algorithms introduced in this section.

a reasonable comparator when, as in the Delicious dataset, there are many moderately popular items. On the other hand, LinUCB-SIN is a competitive baseline when, as in the Last.fm dataset, there are few very popular items. The two scalable variants of GOB.Lin which we empirically analyzed are based on node clustering,[6] and are defined as follows.

**GOB.Lin.MACRO:** GOB.Lin is run on a *weighted* graph whose nodes are the clusters of the original graph. The edges are weighted by the number of inter-cluster edges in the original graph. When all nodes are clustered together, then GOB.Lin.MACRO recovers the baseline LinUCB-SIN as a special case. In order to strike a good trade-off between the speed of the algorithms and the loss of information resulting from clustering, we tested three different cluster sizes: 50, 100, and 200. Our plots refer to the best performing choice.

**GOB.Lin.BLOCK:** GOB.Lin is run on a disconnected graph whose connected components are the clusters. This makes $A_\otimes$ and $M_t$ (Figure 2) block-diagonal matrices. When each node is clustered individually, then GOB.Lin.BLOCK recovers the baseline LinUCB-IND as a special case. Similar to GOB.Lin.MACRO, in order to trade-off running time and cluster sizes, we tested three different cluster sizes (5, 10, and 20), and report only on the best performing choice.

As the running time of GOB.Lin scales quadratically with the number of nodes, the computational savings provided by the clustering are also quadratic. Moreover, as we will see in the experiments, the clustering acts as a regularizer, limiting the influence of noise. In all cases, the parameter $\alpha$ in Figures 1 and 2 was selected based on the scale of instance vectors $\bar{x}_t$ and $\widetilde{\phi}_{t,k_t}$, respectively, and tuned across appropriate ranges. Table 2 and Figure 4 show the cumulative reward for each algorithm, as compared ("normalized") to that of the random predictor, that is $\sum_t (a_t - \bar{a}_t)$, where $a_t$ is the payoff obtained by the algorithm and $\bar{a}_t$ is the payoff obtained by the random predictor, i.e., the average payoff over the context vectors available at time $t$. Table 2 (synthetic datasets) shows that GOB.Lin and LinUCB-SIN are more robust to payoff noise than LinUCB-IND. Clearly, LinUCB-SIN is also unaffected by graph noise, but it never outperforms GOB.Lin. When the payoff noise is low and the graph noise grows GOB.Lin's performance tends to degrade. Figure 4 reports the results on the two real-world datasets. Notice that GOB.Lin and its variants always outperform the baselines (not relying on graphical information) on both datasets. As expected, GOB.Lin.MACRO works best on Last.fm, where many users gave positive payoffs to the same few items. Hence, macro nodes apparently help GOB.Lin.MACRO to perform better than its corresponding baseline LinUCB-SIN. In fact, GOB.Lin.MACRO also outperforms GOB.Lin, thus showing the regularization effect of using macro nodes. On Delicious, where we have many moderately popular items, GOB.Lin.BLOCK tends to perform best, GOB.Lin being the runner-up. As expected, LinUCB-IND works better than LinUCB-SIN, since the former is clearly more prone to personalize item recommendation than the latter. Future work will consider experiments against different methods for sharing contextual and feedback information in a set of users, such as the feature hashing technique of [22].

**Acknowledgments**
NCB and GZ gratefully acknowledge partial support by MIUR (project ARS TechnoMedia, PRIN 2010-2011, contract no. 2010N5K7EB-003). We thank the Laboratory for Web Algorithmics at Dept. of Computer Science of University of Milan.

## Footnotes

[1] Formally, $i_t$ and $C_{i_t}$ can be arbitrary (measurable) functions of past rewards $a_1, \ldots, a_{t-1}$, indices $i_1, \ldots, i_{t-1}$, and sets $C_{i_1}, \ldots, C_{i_{t-1}}$.

[2] The Kronecker product between two matrices $M \in \mathbb{R}^{m \times n}$ and $N \in \mathbb{R}^{q \times r}$ is the block matrix $M \otimes N$ of dimension $mq \times nr$ whose block on row $i$ and column $j$ is the $q \times r$ matrix $M_{i,j}N$.

[3] Datasets and their full descriptions are available at `www.grouplens.org/node/462`.

[4] In the context of recommender systems, these two datasets may be seen as representatives of two "markets" whose products have significantly different market shares (the well-known dichotomy of hit vs. niche products). Niche product markets give rise to power laws in user preference statistics (as in the blue plot of Figure 3).

[5] We did not repeat the same operation on Last.fm because this dataset was already extremely sparse.

[6] We used the freely available Graclus (see e.g. [11]) graph clustering tool with normalized cut, zero local search steps, and no spectral clustering options.

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
