[Supplementary Material]

# Supplementary material to
# "A Gang of Bandits"

**Nicolò Cesa-Bianchi**
Università degli Studi di Milano, Italy
nicolo.cesa-bianchi@unimi.it

**Claudio Gentile**
University of Insubria, Italy
claudio.gentile@uninsubria.it

**Giovanni Zappella**
Università degli Studi di Milano, Italy
giovanni.zappella@unimi.it

## A   Appendix

This appendix contains the proof of Theorem 1.

*Proof.* Recall that

$$\widetilde{U} = A_\otimes^{1/2} U \qquad \text{where} \qquad U = (\boldsymbol{u}_1^\top, \boldsymbol{u}_2^\top, \ldots, \boldsymbol{u}_n^\top)^\top \in \mathbb{R}^{dn} \ .$$

Let then $t$ be a fixed time step, and introduce the following shorthand notation:

$$\boldsymbol{x}_t^* = \operatorname*{argmax}_{k=1,\ldots,c_t} \boldsymbol{u}_{i_t}^\top \boldsymbol{x}_{t,k} \qquad \text{and} \qquad \widetilde{\boldsymbol{\phi}}_t^* = \operatorname*{argmax}_{k=1,\ldots,c_t} \widetilde{U}^\top \widetilde{\boldsymbol{\phi}}_{t,k} \ .$$

Notice that, for any $k$ we have

$$\widetilde{U}^\top \widetilde{\boldsymbol{\phi}}_{t,k} = U^\top A_\otimes^{1/2} A_\otimes^{-1/2} \boldsymbol{\phi}_{i_t}(\boldsymbol{x}_{t,k}) = U^\top \boldsymbol{\phi}_{i_t}(\boldsymbol{x}_{t,k}) = \boldsymbol{u}_{i_t}^\top \boldsymbol{x}_{t,k} \ .$$

Hence we decompose the time-$t$ regret $r_t$ as follows:

$$
\begin{aligned}
r_t &= \boldsymbol{u}_{i_t}^\top \boldsymbol{x}_t^* - \boldsymbol{u}_{i_t}^\top \boldsymbol{x}_{t,k_t} \\
&= \widetilde{U}^\top \widetilde{\boldsymbol{\phi}}_t^* - \widetilde{U}^\top \widetilde{\boldsymbol{\phi}}_{t,k_t} \\
&= \widetilde{U}^\top \widetilde{\boldsymbol{\phi}}_t^* - \boldsymbol{w}_{t-1}^\top \widetilde{\boldsymbol{\phi}}_t^* + \boldsymbol{w}_{t-1}^\top \widetilde{\boldsymbol{\phi}}_t^* + \mathrm{CB}_t(\widetilde{\boldsymbol{\phi}}_t^*) - \mathrm{CB}_t(\widetilde{\boldsymbol{\phi}}_t^*) - \widetilde{U}^\top \widetilde{\boldsymbol{\phi}}_{t,k_t} \\
&\le \widetilde{U}^\top \widetilde{\boldsymbol{\phi}}_t^* - \boldsymbol{w}_{t-1}^\top \widetilde{\boldsymbol{\phi}}_t^* + \boldsymbol{w}_{t-1}^\top \widetilde{\boldsymbol{\phi}}_{t,k_t} + \mathrm{CB}_t(\widetilde{\boldsymbol{\phi}}_{t,k_t}) - \mathrm{CB}_t(\widetilde{\boldsymbol{\phi}}_t^*) - \widetilde{U}^\top \widetilde{\boldsymbol{\phi}}_{t,k_t},
\end{aligned}
$$

the inequality deriving from

$$\boldsymbol{w}_{t-1}^\top \widetilde{\boldsymbol{\phi}}_{t,k_t} + \mathrm{CB}_t(\widetilde{\boldsymbol{\phi}}_{t,k_t}) \ge \boldsymbol{w}_{t-1}^\top \widetilde{\boldsymbol{\phi}}_{t,k} + \mathrm{CB}_t(\widetilde{\boldsymbol{\phi}}_{t,k}), \qquad k = 1, \ldots, c_t.$$

At this point, we rely on [1] (Theorem 2 therein with $\lambda = 1$) to show that

$$\left| \widetilde{U}^\top \widetilde{\boldsymbol{\phi}}_t^* - \boldsymbol{w}_{t-1}^\top \widetilde{\boldsymbol{\phi}}_t^* \right| \le \mathrm{CB}_t(\widetilde{\boldsymbol{\phi}}_t^*) \qquad \text{and} \qquad \left| \boldsymbol{w}_{t-1}^\top \widetilde{\boldsymbol{\phi}}_{t,k_t} - \widetilde{U}^\top \widetilde{\boldsymbol{\phi}}_{t,k_t} \right| \le \mathrm{CB}_t(\widetilde{\boldsymbol{\phi}}_{t,k_t})$$

both hold simultaneously for all $t$ with probability at least $1 - \delta$ over the noise sequence. Hence, with the same probability,

$$r_t \le 2\,\mathrm{CB}_t(\widetilde{\boldsymbol{\phi}}_{t,k_t})$$

holds uniformly over $t$. Thus the cumulative regret $\sum_{t=1}^{T} r_t$ satisfies

$$
\begin{aligned}
\sum_{t=1}^{T} r_t &\leq \sqrt{T \sum_{t=1}^{T} r_t^2} \\
&\leq 2\sqrt{T \sum_{t=1}^{T} \left(\mathrm{CB}_t(\widetilde{\phi}_{t,k_t})\right)^2} \\
&\leq 2\sqrt{T \left(\sigma \sqrt{\ln \frac{|M_T|}{\delta}} + \|\widetilde{U}\|\right)^2 \sum_{t=1}^{T} \widetilde{\phi}_{t,k_t}^\top M_{t-1}^{-1} \widetilde{\phi}_{t,k_t}} \, .
\end{aligned}
$$

Now, using (see, e.g., [2])

$$
\sum_{t=1}^{T} \widetilde{\phi}_{t,k_t}^\top M_{t-1}^{-1} \widetilde{\phi}_{t,k_t} \leq \left(1 + \max_{k=1,\ldots,c_t} \|\widetilde{\phi}_{t,k}\|^2\right) \ln |M_T| \, ,
$$

with

$$
\begin{aligned}
\max_{k=1,\ldots,c_t} \|\widetilde{\phi}_{t,k}\|^2 &= \max_{k=1,\ldots,c_t} \phi_{i_t}(\boldsymbol{x}_{t,k}) A_\otimes^{-1} \phi_{i_t}(\boldsymbol{x}_{t,k}) \\
&\leq \max_{k=1,\ldots,c_t} \|\phi_{i_t}(\boldsymbol{x}_{t,k})\|^2 \\
&= \max_{k=1,\ldots,c_t} \|\boldsymbol{x}_{t,k}\|^2 \\
&\leq B^2 \, ,
\end{aligned}
$$

along with $(a+b)^2 \leq 2a^2 + 2b^2$ applied with $a = \sigma\sqrt{\ln \frac{|M_T|}{\delta}}$ and $b = \|\widetilde{U}\|$ yields

$$
\sum_{t=1}^{T} r_t \leq 2\sqrt{T \left(2\sigma^2 \ln \frac{|M_T|}{\delta} + 2\|\widetilde{U}\|^2\right)(1 + B^2)\ln |M_T|} \, .
$$

Finally, observing that

$$
\|\widetilde{U}\|^2 = U^\top A_\otimes U = L(\boldsymbol{u}_1, \ldots, \boldsymbol{u}_n)
$$

gives the desired bound. □