[Reviews · NeurIPS 2013]

Submitted by Assigned_Reviewer_1

Thanks to your rebuttal, I think I now understand your algorithm, and I think it is correct. But why did you present in Figure 2 algorithm 2 with CB and not TCB? The algorithm with CB does not work, and it is misleading to put CB in Figure 2. I would recommend changing this and putting TCB in the presentation of your algorithm.

Also, please comment on the necessity of knowing L(u_1,...,u_n) (or rather an upper bound on this, and rewrite the Thm with an upper bound since it is not realistic to have truly this quantity available). This assumption is not innocent since it implies that you assume some knowledge on the quality of the graph you have (how it relies to the u_1,..., u_n).


*********************OLD REVIEW without comments on Thm 1


This paper considers an extension of linear bandit algorithms, i.e. linear bandits with graph information. For instance a recommendation system can have access to a social network to which the users belong to, and it is reasonable to assume that users that are linked together might share interests. Each user can be modelled as a linear bandit problem, these bandits being linked together by a graph. The graph represents the level of similitude between the parameters of the bandits.

The authors assume that the graph is known to the algorithm, that at each time t the algorithm receives an user ID and a context, and that it has to recommend an arm according to that. They propose an algorithm, GOB.lin, that solves this problem based on LinUCB. The difference with respect to this last algorithm is that this algorithm exploits the graph information to make use of the similitudes between the users of the recommendation system. The authors provide a bound on the regret of this algorithm, and some numerical experiments that are promising and convincing.


Also, it would be nice if you could add some more intuition on your algorithm. Can it be seen as n bandits where, if some arm i_t is selected at time t, the estimate w_{j,t} of an arm j will be updated as w_{j,t,k} = (1-e(i_t,j)) w_{j,t-1,k} + e(i_t,j) \tilde r_{t},
where \tilde r_t = M_{t-1}^{-1} a_t \tilde \phi_{t,k} is the "classic linear bandit update" on w_{i_t,t}, and e(i_t,j) is between 0 and 1/t, and is a measure of the distance between i_t and j, that depends on i_t and j only and can be easily computed using the graph (e.g. if two nodes are linked e(i_t,j) = 1/t, if there is no path between them, e(i_t,j) = 0, and then anything in between depending on their proximity level?)? This would make your explanations clearer and your algorithm simpler.

Other remarks:
- p2 l89: "....a new model distribuited..."??
- p3 l129: "We model the similarly among users...."??
Summary: After reading the author rebuttal, I now think the proof is correct and the algorithm works - but please change its presentation, you propose an algorithm that does not work as such (Fig 2), and then analyse a modification of this algorithm, that works (see the TCB in Th 1).

I changed completely my score and now recommend acceptance of this paper, that is interesting and on a hot topic. I would however recommend that the authors rewrite their paper in a much clearer way for the final version.

Submitted by Assigned_Reviewer_4

-------Original Comments-------------

This paper provides a UCB-based contextual bandit algorithm that can exploit information given in the form of a graph structure. In particular the authors consider a contextual bandit model with linear payoffs and unknown coefficient vectors that depend on the user. A graph structure on the users is given, and an assumption on the relationship between the graph structure and the user coefficient vectors ensures that the graph provides meaningful information about the coefficient vectors. The algorithm then exploits this information, using the Laplacian matrix of the graph to propagate information gained about one user to other users with similar coefficient vectors. The authors provide both a regret bound and numerical experiments to support their proposed approach.

This is a natural extension of the techniques currently available, and part of an important general direction to integrate bandit ideas in more practical scenarios. Overall the paper is quite well written. However there are some sentences where the English is not quite correct (see below for some of these typos), and I did not find the intuition given in Section 4 to be easy to understand.

--------After Rebuttal--------------

Having discussed this submission in detail with the other reviewers, I have decided to lower my quality score. Initially I thought that my lack of familiarity with graph based methods hindered my understanding of the intuition and choices made in the paper. However, now I am convinced that the intuition is not well given. Also, it is strange to see the implementation of CB, but the analysis of TCB. There is no discussion of whether CB can be analysed or not. While it is true that there is a discrepency between LinUCB and SupLinUCB, the former was studied in detail empirically, and the latter in theory in a separate paper. There is nothing wrong (technically) with the elements of the contribution, I no longer find it to be a strong candidate for acceptance.

-----------Some typos:--------------

129 We model the similary --> We model the similarity
137 That is, despite (1) may contain ... --> That is, although (1) may contain ...
207 but it rather contains graph ... --> but contains the graph ...
371/372 lends itself to be --> turns out to be
Summary: The paper provides an interesting and natural extension of contextual bandits to incorporate other information available in the form of a graph, a subject of interest to the machine learning community. However the work is not surprising, and there are several parts of the work, such as the intuition, which are not well executed.

Submitted by Assigned_Reviewer_5

This paper considers a setting where several contextual linear bandits are connected by the underlying graph. The assumptions is the weight vectors of the respective nodes are smooth over the graph (are close for neighbors). This appears to a be a good and rich enough model for recommendations in social networks.

The algorithm runs classical contextual linear bandits on each node (which node is being tried is determined by nature), but then shares this information with other nodes as well so they also update their model estimates and confidence widths appropriately via Laplacian.

Unfortunately, the regret analysis provided is disappointing. The authors in fact provide the analysis for a different algorithm than they propose and evaluate in the experiments. This could be perhaps acceptable but I found no reason whatsoever over for this. There is no discussion,
how CB vs. TCB would change the algorithm, its evaluation and the deployment. At least the difference between TCB and CB should be discussed better than one could use the results from [1]. It is therefore possible, that the TCB-kind algorithm does not do well in the experiments or that for CB-kind it is not possible to provide learning guarantees.
Maybe it would be better just to not include such analysis.

Regarding the current analysis, it is not clear how big the terms ln|M_t| and 2L can be. How do they behave with T and n? This is needed in order to get some insight from the provided regret bound. And how does the graph noise influence these quantities?

Furthermore, typically [4,9] the analyses for linear bandits give at least a guideline to set the exploration parameter \alpha. Unfortunately, we do get this from the analysis provided.
What is the dependence of the upper bound on alpha anyway? Is it hidden in some of the quantities mentioned above?

The experiment on the other hand are treated very well. The descriptions are very detailed and well explained. It is also great that the authors performed the experiments on real data and described the practical details to make the experiments reproducible.

Finally, L186 - L190 The paper mentions that through graph, the reward influences other nodes in the "lesser extent", "somehow informatively" and "similar updates" are desired. However I do not see a parameter for "somehow" or "similar" in the algorithm. Could that be done by regularizing L by a multiple of I instead of I. In the experimental section, the authors mention that the clustering acts as a regularizer. Would not regularizing the Laplacian instead be
a more principled way to achieve this regularization?

Other comments
- Why it is L(u1,...un) in the regret bound and not the regularized version,
since the Laplacian is regularized in the algorithm?

After the rebuttal: While this is a good work, I believe that including the theoretical analysis of a changed algorithm with not entirely clear relationship between the two, makes it confusing. I would increase my score if the theoretical analysis was taken away.
Summary: This paper extends the contextual bandits to the setting where the different bandits share their weight-vector via provided graph (useful in the social network setting). Very good experiments, but the analysis provided is disappointing and does not appear to be useful.
Author Feedback

Author rebuttal: Reviewer 1:
* On the correctness of the analysis ("there are two nodes that are connected ... very different u" + "I do not see why Thm 2 in [1] would apply").
Admittedly, our presentation leaves out many technical details. Yet, our analysis is a sound combination of previous results. In fact:
1) Our alg simply runs linear least squares in a RKHS with vectors \tilde{\Phi} as elements, and whose inner product is based on the inverse Laplacian of the graph.
2) Thm 2 of [1] is a general result applying to linear least squares and subgaussian payoffs a_t. This applies to our setting because inner products in the RKHS and the original space are matching (Line 037 in the suppl. material).
3) The reviewer is right in that the presence of two connected nodes with very different u's hurts both the alg and the bound: the resulting bias contributes to increasing the quantity L(u_1, ... u_n) in Line 233. Note that, in Line 244, L(u_1, ... u_n)*T is under the square root, so convergence still takes place. As an example, let G have only two nodes 1 and 2 along with edge (1,2), and let the sequence i_t alternate between 1 and 2. Because we are using a regularized Laplacian I + L, least squares performs a "full" update on node 1 when i_t = 1, a "full" update on node 2 when i_t = 2, but only "fractional" updates on node 2 when i_t = 1 and on node 1 when i_t = 2. To see it, pls check the resulting vectors \tilde{\Phi} (but see also below). Hence, the bias introduced by updating "the wrong way" at node 2 with payoffs coming from node 1 is offset by updating "the right way" at node 2 with payoffs coming from node 2. Same thing happens at node 1.

We will certainly add more explainations about why Thm 2 in [1] applies to our setting. Pls see also the response to Reviewer 5.

* More intuition on the algorithm.
Our alg is multitask linear least squares, each node being a task, and task connectedness (closeness of u vectors) corresponding to graph edges. For instance, if G is the n-clique, then vectors \Phi_{i_t}(x_{t,k}) in Fig 2 are sparse block vectors, and vectors \tilde{\Phi}_{t,k} are dense vectors, where in block i_t vector x_{t,k} gets multiplied by some fraction of unity and, *in all other* blocks, vector x_{t,k} still occurs, but multiplied by a *smaller* fraction of unity. This amounts to performing an update on all nodes of G based on the payoff observed at node i_t, but weighting them differently. In this specific example, the update at node i_t is worth roughly \sqrt{n} times the update made at any other node.

Reviewer 1 has the right intuition about how the alg works. However, the update cannot be written down the way s/he suggests, and the "distance" alluded at is a spectral distance (called the resistance distance) which can be read off from the inverse Laplacian matrix of G.

We will certainly add more on the way Laplacian-based regularization operates (e.g., in the vein of the above examples), though much of this can already be found in the literature on multitask learning (e.g., [8], Sect. 3.3 and Sect. 4.2).

* Other remarks.
We'll fix the typos, thanks.



Reviewer 5:
* On the mismatch CB vs. TCB in analysis and experiments + theoretical guidelines for setting alpha.
We did not use TCB in our experiments because the upper confidence bound TCB comes from too conservative an analysis to be applied *as is* in practice, and TCB depends anyway on *unknown* quantities (like L). Observe that even LinUCB in [9] has this issue when tuning alpha: the theoretical guidelines for setting alpha therein apply at the cost of assuming a unit norm for the comparison vector \theta^*. Yet, our unknown parameter L (Line 233) also depends on the distances among comparison vectors, which is specific to this more general social network scenario.
We would like to stress that CB and TCB share *a very similar* dependence on time: the quantity ln|M_t| is logarithmic in t, but its dependence on n heavily depends on the graph G. Notice the following two extreme cases:
1) When G has no edges then trace(M_t) = trace(I) + t = nd + t, hence ln|M_t| \leq dn*log(1+t/(dn)).
2) When G is the complete graph then trace(M_t) = trace(I) + 2t/(n+1) = nd + 2t/(n+1), hence ln|M_t| \leq dn*log(1+2t/(dn(n+1))).
The precise behavior of ln|M_t| (one that would ensure a significant advantage in practice) depends on the actual interplay between the data and the graph, so that the above linear dependence on dn is really a coarse upper bound. In our experiments, we simply incorporated all this knowledge in the single tunable parameter alpha. To clarify things, we will add to the paper a discussion along the above lines.

* "not clear how big the terms ln|M_t| and 2L can be".
See above for term ln|M_t|. As for L (see Line 233), this depends on the distance among the unknown u_i in G, pls see discussion surrounding Line 134. Graph noise is indeed an important issue that we tackled through node clustering -- see below.

* Dependence of upper bound on alpha.
Our theoretical analysis is not parameterized by alpha. The analysis only applies to the case when alpha in Fig. 2 is such that CB = TCB.

* On the influence of the graph in the updates and "regularized Laplacian" + clustering as a regularization.
The reviewer is right in that cI+L is a way of regularizing L which would yield regret bounds (the optimal tuning of c depending on unknown quantities). Though very interesting (and capable of reducing graph noise), this regularization does not ensure the computational advantages achieved by clustering nodes. The two regularizations are indeed quite different, and one cannot simulate one through the other.

* "Other comments".
L(u_1 ... u_n) is in fact the scalar ||\tilde{U}||^2 (where \tilde{U} is defined in Line 030 in the suppl. material), which is equal to what is in Line 233. Notice that L(u_1 ... u_n) incorporates the regularizating term I in I+L through the dependence on \sum_{i \in V} ||u_i||^2.